# Quantitative Trait Locus Mapping of Marsh Spot Disease Resistance in Cranberry Common Bean (*Phaseolus vulgaris* L.)

**DOI:** 10.3390/ijms23147639

**Published:** 2022-07-11

**Authors:** Bosen Jia, Robert L. Conner, Waldo C. Penner, Chunfang Zheng, Sylvie Cloutier, Anfu Hou, Xuhua Xia, Frank M. You

**Affiliations:** 1Ottawa Research and Development Centre, Agriculture and Agri-Food Canada, Ottawa, ON K1A 0C6, Canada; bosen.jia@agr.gc.ca (B.J.); chunfang.zheng@agr.gc.ca (C.Z.); sylviej.cloutier@agr.gc.ca (S.C.); 2Department of Biology, University of Ottawa, 30 Marie Curie, Ottawa, ON K1N 6N5, Canada; xuhua.xia@uottawa.ca; 3Morden Research and Development Centre, Agriculture and Agri-Food Canada, Morden, MB R6M 1Y5, Canada; robert.conner150@gmail.com (R.L.C.); waldo.penner@agr.gc.ca (W.C.P.)

**Keywords:** marsh spot disease, cranberry common bean, QTL mapping, genotyping by sequencing (GBS), single nucleotide polymorphisms (SNPs), genome-wide association study (GWAS), *Phaseolus vulgaris*

## Abstract

Common bean (*Phaseolus vulgaris* L.) is a food crop that is an important source of dietary proteins and carbohydrates. Marsh spot is a physiological disorder that diminishes seed quality in beans. Prior research suggested that this disease is likely caused by manganese (Mn) deficiency during seed development and that marsh spot resistance is controlled by at least four genes. In this study, genetic mapping was performed to identify quantitative trait loci (QTL) and the potential candidate genes associated with marsh spot resistance. All 138 recombinant inbred lines (RILs) from a bi-parental population were evaluated for marsh spot resistance during five years from 2015 to 2019 in sandy and heavy clay soils in Morden, Manitoba, Canada. The RILs were sequenced using a genotyping by sequencing approach. A total of 52,676 single nucleotide polymorphisms (SNPs) were identified and filtered to generate a high-quality set of 2066 SNPs for QTL mapping. A genetic map based on 1273 SNP markers distributed on 11 chromosomes and covering 1599 cm was constructed. A total of 12 stable and 4 environment-specific QTL were identified using additive effect models, and an additional two epistatic QTL interacting with two of the 16 QTL were identified using an epistasis model. Genome-wide scans of the candidate genes identified 13 metal transport-related candidate genes co-locating within six QTL regions. In particular, two QTL (*QTL.3.1* and *QTL.3.2*) with the highest *R*^2^ values (21.8% and 24.5%, respectively) harbored several metal transport genes *Phvul.003G086300*, *Phvul.003G092500*, *Phvul.003G104900*, *Phvul.003G099700*, and *Phvul.003G108900* in a large genomic region of 16.8–27.5 Mb on chromosome 3. These results advance the current understanding of the genetic mechanisms of marsh spot resistance in cranberry common bean and provide new genomic resources for use in genomics-assisted breeding and for candidate gene isolation and functional characterization.

## 1. Introduction

Common bean (*Phaseolus vulgaris* L., 2n = 2x = 22) is a widely grown grain legume crop planted in Canada with areas up to 160,000 ha and dry seed production up to 316,800 Mt [1]. As reported by the FAO (Food and Agriculture Organization), common bean global production reached 28.9 million tons within 33.1 million ha around the world in 2019. Over half of global production was shared by Asians [2]. It is not only a crucial crop for food security, but it is also highly nutritious, meeting human nutrition requirements for proteins, vitamins, minerals, carbohydrates, and other nutrients. The common bean is a high-quality and low-cost source of proteins, especially valuable for developing countries.

One of the most important goals in common bean breeding is yield. High-yield cultivars are more likely to be selected during long-time domestication, and recent research also focused on improving bean yield [3,4,5]. With the help of new cultivars and technologies, the average common bean yield has achieved a remarkable increase of approximately 420 kg per ha for over 40 years from 1961–2016 [6].

Other traits such as micronutrient contents or resistance to abiotic and biotic stresses are also crucial to sustaining common bean yield and quality. Marsh spot disease is a physiological disorder that affects both seed yield and quality of pulse crops, primarily in peas [7,8,9,10,11,12,13] and beans [7,10,14]. Marsh spot, first reported in 1933 on peas, is characterized by a brown lesion in the flat inner surface of one or both cotyledons that is sometimes accompanied by partial or entire necrosis of the plumule [15]. In bean seeds affected by marsh spots, discolored lesions usually occur at the center of the adaxial surface of each cotyledon within the seed [16]. These spots may be caused by cell death underneath the cotyledon’s skin [13]. A brown substance replaces the starch originally stored in those cells and, eventually, seed staining becomes visible [13].

Although marsh spot disease was discovered in 1933, its inheritance has not been thoroughly studied because nutrient deficiencies do not generally get the same attention as pathogenic disorders. While some chemical treatments have been used to reduce the incidence and severity of marsh spot [17,18], genetic improvement remains the most efficient and environment-friendly approach. To understand the genetic mechanism of marsh spot disease, an F_2:7_ population consisting of 138 recombinant inbreeding lines (RILs) from a cross between the susceptible cultivar ‘Messina’ and the resistant cultivar ‘Cran09’ was evaluated for the presence and extent of marsh spot lesions over a five-year period on two soils: sand and heavy clay [19]. Both the marsh spot incidence (MSI) and resistance index (MSRI) were used to estimate the disease severity. The highly correlated MSRI and MSI showed high broad-sense heritability values (*H*^2^) of 86.5% and 83.2%, respectively [19]. There was no significant difference between the two soil types across five years for MSRI and MSI. The joint segregation analysis of the phenotypic data of marsh spot reactions revealed that at least four major genes controlled marsh spot resistance [19].

Previous studies indicated that marsh spot is caused by a manganese (Mn) deficiency in pea [8,9,11,12,16,20,21,22]. However, only a few physiological and genetic experiments have been performed on marsh spot in cranberry common bean [19,23]. In order to examine the effect of Mn on marsh spot disease, the Mn concentration in the soil of the used experimental fields was tested [19]. The Mn concentrations in the soil of the fields in the first three years (2015–2017) were much higher than those in the last two years (2018–2019); at the same time, significantly lower MSRI and MSI in the first three years than in the last two years were also observed, indirectly supporting the relationship between Mn content and marsh spot disease [19].

Zinc transporters (ZIP), vacuolar iron transporter (VIT), natural resistance-associated macrophage protein (NRAMP) and cation flux (CAX) were considered as important protein families in Mn transportation in plants and were co-located with identified QTL in *Phaseolus vulgaris*, *Lotus japonicas*, *Lens culinaris*, *Brassica napus*, *Brassica rapa*, *Hordeum vulgare*, and *Oryza sativa* [16]. The ZIP transporters commonly occur in bacteria, fungi, plants, and animals and are considered to be associated with Fe^2+^, Zn^2+^, Cd^2+^, Co^2+^, Cu^2+^, and Mn^2+^ transport. They have eight transmembrane domains (TMD) with extracellular N- and C-termini and a cytosolic histidine-rich loop [24,25]. Yellow stripe-like (YSL) transporters are linked to the oligopeptide transporter (OPT) family and occur only in plants, bacteria, fungi, and archaea. Members of the YSL family were predicted to transport metals (Mn^2+^, Zn^2+^, Cu^2+^, Ni^2+^, Cd^2+^, Fe^2+^) complexed to non-proteinogenic amino acids, such as nicotinamide (NA) or Phyto-siderophores [20,26]. NRAMP protein families are members of the major proteins implicated in Mn transportation from the root to the stem [27,28]. The CAX family mainly regulates the influx of cations into the vacuole. Its members are metal transporters that arbitrate the influx of cations into the vacuole [29,30].

Here, we report on the quantitative trait loci (QTL) associated with marsh spot resistance and on the putative candidate genes with a goal to assist in the development of diagnostic markers for marker-assisted breeding and to provide genomics resources towards the cloning of the causal genetic features of marsh spot in beans.

## 2. Results

### 2.1. SNP Identification

A total of 13,064,398 paired-end genotyping by sequencing (GBS) reads corresponding to 196 Mb were generated from the sequencing of the 138 recombinant inbred lines (RILs). Considering a genome size of 473 Mb [31], the average genome coverage was 3.65X per line, ranging from 0.12X to 11.97X. To identify the parental origin of the variants identified in the RILs, the two parents were sequenced at a high coverage depth of 32.89X for Cran09 and 29.45X for Messina. An average of 78.56% of the reads of the RILs were aligned to the Andean type common bean genome G19833 reference genome (V2.1) [31], ranging from 66.54% to 82.70% (Appendix A).

A total of 54,620 single nucleotide polymorphisms (SNPs) were identified by aligning GBS reads of the 138 RILs to the reference genome (V2.1) [31]. Filtering for minor allele frequency (MAF) > 0.01 and call rate > 20% yielded a total of 2066 SNPs. In addition, eight SNPs mapped to small scaffolds and were removed (Appendix A). The SNPs were distributed across the whole genome, with an average of 188 SNPs per chromosome (Chr) (Appendix A). Some regions on Chr 2, 3, 4, 6 displayed high-density SNP regions (Appendix A). Among the 2058 SNPs, 1863 SNPs were polymorphic between parents (Cran09 and Messina), and 195 SNPs had no call in one of the parents (Table 1). Then, 785 SNPs that had significant segregation distortion at a 0.05 probability level were eliminated. Finally, 1273 SNPs were further imputed and used for linkage map construction (Figure 1).

### 2.2. Genetic Map

A genetic map of the 11 linkage groups or chromosomes was constructed containing 1273 SNP markers ranging from 9 on Chr 8 to 360 on Chr 5. Most SNPs that were identified on the same chromosomes on the reference sequence were grouped into the same linkage groups (Appendix A) and showed consistent orders in the physical chromosomes (Appendix A). The map consisted of 423 recombination intervals with a total length of 1599 cm and an average interval of 3.78 cm (Table 1). Since only nine markers were retained on Chr 8 after removing SNPs of significant segregation distortion, a large average interval (36.11 cm) between markers was obtained.

### 2.3. Genomic Heritability

The genomic heritability (*h*^2^) of common bean resistance to marsh spot was estimated for MSRI using the genetic additive variance of all SNPs and phenotypes by GBLUP. The *h*^2^ estimates ranged from 12.07% to 55.91% in all 18 datasets with the highest *h*^2^ (55.91%) originating from the overall mean dataset of MSRI (Table 2).

### 2.4. Mapping of Additive QTL

Using two genetic map-based statistical models (ICIM-ADD and GCIM) and the haplotype block-based genome-wide association study (GWAS) model RTM-GWAS, a total of 18 QTL were identified from 18 phenotypic datasets. The QTL identified using different models were grouped into single QTL because they co-located on chromosomes or were within the same haplotype block. To validate the QTL identified by the different statistical models and from different phenotypic datasets (environments), we performed single factor (alleles) ANOVA for each identified QTL using the 18 phenotypic datasets. Two of the 18 QTL had no significant allelic differences in QTL effects in >15 datasets and were removed. The 12 stable QTL presented significant QTL effects in most of the phenotypic datasets (>10) with the mean *R*^2^ ranging from 6.81% (*QTL.6.1*) to 24.52% (*QTL.3.1*), whereas the remaining four QTL (*QTL.1.1*, *QTL.5.1*, *QTL.6.2* and *QTL.9.1*) explained 5.9–7.8% of phenotypic variation in three to five phenotypic datasets, indicative of environment-specific features (Table 3 and Appendix A, Figure 2).

Of the sixteen QTL, one was located on Chr 1, six on Chr 2, two on Chr 3, four on Chr 5, two on Chr 6, and one on Chr 9 (Table 3). One QTL was identified by a single SNP, or a quantitative trait nucleotide (QTN). In all sixteen QTL, two QTL were detected by three models, seven QTL by two models, and seven QTL by only one model. The LOD value of a QTL represents its significance extent. The LOD values for QTL identified from ICIM-ADD and GCIM varied from 3.14 to 7.58. Thirteen out of sixteen QTL had relatively high absolute values of additive effects (≥0.1) ranging from 0.1 to 0.57 (Table 3).

### 2.5. Mapping of Epistatic QTL

The additive-epistasis model ICIM-EPI was used to detect interactions among QTL. A total of three QTL pairs with significant epistatic effects were identified, involving two additive QTL identified using additive models: *QTL.2.3* and *QTL.5.4* (Table 4, Figure 3). *QTL.5.4* significantly interacted with two additional QTL, *QTL.2.7* and *QTL.2.8* identified by ICIM-EPI, while *QTL.2.3* also interacted with *QTL.2.8* (Table 4). The LOD values varied from 6.21 to 7.20. The additive effects of the two QTL *QTL.2.3* and *QTL.5.4* were −0.06 and −0.05, respectively, and those of the two interacting QTL *QTL.2.7* and *QTL.2.8* were the same value of 0.04. The epistatic effects of the three pairs of QTL ranged from −0.10 to −0.11. The mean *R*^2^ of QTL ranged from 16.31% to 30.69%, and the highest average *R*^2^ of 30.69% was obtained from the QTL pair *QTL.2.9* and *QTL.5.4*.

Despite the significant interactions between pairs of QTL, on average, the number of favorable alleles of individuals tended to be positively correlated with MSRI (Appendix A).

### 2.6. Contribution of All Detected QTL to Marsh Spot Resistance

In examining only additive effects, multiple linear regression models of all 16 QTL for each of 18 phenotypic datasets were constructed to calculate the overall contribution of all QTL to the phenotype variation. The *R*^2^ of the model represents the portion of the phenotypic variation explained by all 16 QTL. The *R*^2^ estimates of the 18 regression models ranged from 46.08% (S2019) to 75.37% (overall dataset), with a mean *R*^2^ of 61.98% (Figure 4). However, when both additive and epistatic effects of the QTL were considered, i.e., the additional two QTL that were influenced by the significant epistatic effects of three of the 16 QTL, the *R*^2^ estimates of the models increased and ranged from 56.21% to 81.87% with a mean *R*^2^ of 69.64% (Figure 4).

The relative contribution (RC) of each QTL to MSRI values estimated in the datasets is listed in Appendix A. *QTL.5.4* which also had a significant epistatic effect with *QTL.2.7* and *QTL.2.8* had the largest RC (13.48%), followed by *QTL.2.1* (10.4%), *QTL.5.3* (9.44%), *QTL.2.6* (8.9%), *QTL.3.2* (8.26%), *QTL.3.1* (8.14%). The remaining QTL had relatively low RCs (ranging from 0.3 to 6.79%). However, due to partial correlation among QTL, the RC and mean *R*^2^ values of QTL were not always consistent (Figure 5). For example, *QTL.3.1* and *QTL.3.2* had the highest mean *R*^2^ values (24.5% and 21.8%, respectively); however, their mean RC values over the 18 datasets were not the highest (7.91% and 7.93%, respectively) (Figure 5, Appendix A).

### 2.7. Favorable Alleles of QTL in RILs

The number of favorable alleles of the 16 additive QTL (Table 3) in each RIL is illustrated in Figure 6. The number of favorable alleles was highly correlated with MSRI values (*R*^2^ = 72.48%) (Figure 7). To further validate this relationship, the overall dataset of the 15 most resistant lines (0.01 ± 0.01 of MSRI) and the 15 most susceptible lines (0.66 ± 0.19 of MSRI) was extracted. The number of favorable alleles of the 15 most resistant lines (11.5 ± 1.8) was significantly higher than that of the 15 most susceptible lines (3.4 ± 1.6) (Figure 8). The resistant parent Cran09 and susceptible parent Messina had 12 and 2 favorable alleles for the 16 QTL, respectively.

The number of favorable alleles for all three pairs of epistatic QTL identified by ICIM-EPI were also counted in the RILs. Interestingly, a significant correlation between the number of favorable alleles and MSRI was still observed, indicating that the additive effects existed despite the significant epistatic effects of these QTL (Appendix A).

### 2.8. Candidate Genes of Major QTL

Since marsh spot disease is most likely caused by Mn deficiency, all potential candidate genes associated with Mn deficiency and Mn content in plants were screened [16]. A list of 151 annotated genes likely related to Mn deficiency or Mn content were identified from other plant species, such as *Arabidopsis*, rice (*Oryza sativa*) and barley (*Hordeum vulgare*) and mapped to the common bean reference genome. Then, a window of upstream and downstream 100 Kb flanking each QTL region was scanned to identify QTL harboring such genes. Six of the QTL co-located with a total of 13 genes (Table 5). Among them, four QTL (*QTL.1.1*, *QTL.3.1*, *QTL.3.2*, and *QTL.5.2*) harbored eight genes encoding heavy metal transport/detoxification superfamily protein. In particular, two QTL (*QTL.3.1* and *QTL.3.2*) which explained the highest phenotypic variation of 12.2–24.5% harbored five metal transport genes *Phvul.003G086300*, *Phvul.003G092500*, *Phvul.003G104900*, *Phvul.003G099700*, and *Phvul.003G108900* in a large genomic region of 16.8–27.5 Mb on Chr 3. Other gene families, including one zinc transporter (ZIP), ZIP metal ion transporter, cation efflux family, natural resistance-associated macrophage protein (NRAMP), and manganese tracking factor for mitochondrial SOD2 were also co-located to *QTL.2.3*, *QTL.5.2*, or *QTL.9.1* (Table 5). These genes are responsible for Mn transporter, metals homeostasis, and detoxification in plants and are very likely to be causal genes controlling marsh spot.

## 3. Discussion

The previous investigation revealed the inheritance of marsh spot resistance in cranberry common beans was likely controlled by at least four genes with additive and epistatic effects [19]. Using joint segregation analysis (JSA) [33,34], the marsh spot phenotypic data of the RIL population best fitted a genetic model with four major genes with additive and interaction effects. The estimated epistatic effects were even larger than additive effects. In this study, using the same population and phenotypic datasets, we identified 16 additive and three pairs of epistatic QTL, validating and confirming that marsh spot resistance is a quantitatively inherited trait controlled by at least four genes. Due to the theoretical limitation of the maximum gene number of the JSA genetic models, a maximum of four major genes can be estimated [33]. In addition, the current version of JSA can only estimate major genes without minor gene effects; thus, the number of major genes was underestimated and the effects from minor genes were ignored. The current study further validated the previous results, extended the discovery, and proposed candidate genes for the QTL that support the role of Mn in marsh spot disease [19].

### 3.1. Genomic Heritability and Contribution of QTL to Marsh Spot Resistance

To understand the overall contribution of the identified QTL, multiple linear regression models of all QTL on each of the 18 phenotypic datasets were constructed. The average *R*^2^ of the models with both additive and epistatic QTL was greater than that of the models with only additive QTL, indicating that the additional epistatic QTL were also useful for improving marsh spot resistance despite some negative interaction between some epistatic QTL (Table 4).

The lowest *R*^2^ values were always obtained from the phenotypic data sets of single years with one soil type (2019/Sandy soil and 2018/Heavy clay soil). The *R*^2^ values of the mean value datasets (such as overall means, means of heavy clay over five years and means of sandy soil over five years) were greater than those from the single environments (single year and single soil type), showing the strong environmental effects on QTL. Of all identified QTL, most were detected from the overall mean dataset or several other mean datasets. Thus, the phenotypic data over multiple years and/or multiple locations helps to identify stable QTL [35].

In the overall mean dataset, genomic heritability (*h*^2^) was estimated to be 55.91%, indicating that the 1273 SNPs identified in the RIL population explained more than half of the phenotypic variance. The missing proportion of the phenotypic variation could be the result of the non-additive effects of SNP markers or missed SNPs in marker-poor regions [36]. The *h*^2^ estimates also varied from different phenotypic datasets. For example, the *h*^2^ was extremely low (12.07%) in the 2019/sandy soil dataset (S2019). MSRI observations in S2019 were significantly lower than those of other datasets possibly due to the higher concentration of Mn in the sandy soil field in 2019. Thus, the estimation of *h*^2^ is possibly affected by the interaction between genetic background and environment [37].

The overall *R*^2^ values of all the identified QTL estimated in the regression models (75.37%) were higher than the *h*^2^ estimates (55.91%) (Table 2). The difference between the two estimates may be because different statistical models were used but this result implied that most of the QTL associated with marsh spot resistance existing in this RIL population may have already been identified using a combination of different statistical models.

### 3.2. Statistical Models for QTL Identification and QTL Validation

Each statistical model for QTL mapping has its own advantages and limitations. The simultaneous utilization of multiple models would be a reasonable and practical strategy to make full use of their merits and overcome potential disadvantages. Several statistical models have been developed for QTL mapping in bi-parental populations, involving linkage map-based models and GWAS models [35,38,39,40]. In this study, four statistical models were used to identify QTL, including three linkage map-based models (i.e., ICIM-ADD, ICIM-EPI and GCIM) and one haplotype block-based GWAS model (i.e., RTM-GWAS). Unlike traditional interval mapping (IM) and composite interval mapping (CIM) models, the ICIM model can control polygenetic background through a prerequisite selection of markers in QTL mapping. Those polygenes with large and moderate effects were well controlled to reduce the rate of false positives [41,42]. GCIM provides a new method to control polygenetic background by estimating polygenetic variance in GWAS, which can control the background of polygenes with large, moderate and small effects. Compared with ICIM models, GCIM outperformed ICIM in small effect QTL detection. However, in some cases in the GCIM model, several peaks around one QTL could be identified at the same time, thus the true QTL was difficult to define. For example, four QTL, *QTL.2.3*, *QTL2.4*, *QTL.2.5* and *QTL.2.6* neighboured each other on the genetic map and on the chromosome in terms of their physical chromosomal locations (Table 3). They spanned a genomic region of 34.0–38.4 Mb or 128.9–188.3 cm on the genetic map. RTM-GWAS first generates and groups SNPs into LD blocks, and then QTL mapping is performed based on these LD blocks, through a process called SNPLDBs [43]. The employment of LD blocks as markers can notably decrease the possibility of false positives during multiple hypotheses in the GWAS model [40]. In this study, all 16 additive QTL were identified using three genetic map-based models (Table 3), demonstrating their detection power in QTL mapping [44,45,46], whereas six of them were validated by RTM-GWAS. These results indicate that the value of using multiple models that combine the genetic map-based and GWAS models facilitates the detection of QTL with small effects and the validation of QTL.

Further validation of the QTL can also be achieved through the use of linear regression models as performed herein where significant correlations between QTL alleles and phenotypes of RILs were shown. Although significant correlations were confirmed for most of the putative QTL identified by statistical models, two of the 18 original QTL did not pass the significance test and were declared false positive QTL.

While all statistical models may result in some false positive QTL, the use of multiple QTL models and other validation methods such as the linear regression models can be capitalized upon to identify them.

### 3.3. Additive/Epistatic QTL and Genomics-Assisted Selection

With the development of genotyping technologies, genomics-assisted selection such as marker-assisted selection (MAS) and genomic selection (GS) has been widely used for the selection of traits controlled by major genes or polygenes in many crops including the common bean [47,48,49,50]. MAS and GS aim at predicting the phenotypes of individuals based on the use of known molecular marker information without expensive or time-consuming phenotyping of the individuals. MAS tends to select superior lines through major genes or large-effect QTL, while GS utilizes high-density genome-wide markers or QTL to predict the performance of individuals.

In this study, a total of 16 additives and three pairs of epistatic QTL have been identified. These QTL explained most phenotypic variations for marsh spot resistance. The accumulation of favorable alleles in RILs via the hybridization of two parents and recombination has greatly improved common bean resistance to marsh spot. The most resistant RILs had significantly greater favorable additive alleles than the most susceptible RILs (Figure 8B). The number of favorable alleles in a RIL had a significantly positive correlation with marsh spot resistance (i.e., negative correlation with MSRI) (Figure 7), showing a significant increase in the number of favorable alleles from susceptible to resistance lines (Figure 6). MAS or GS are both effective in pyramid favorable alleles of QTL to develop future resistant cultivars in plant breeding. The lines containing more favorable alleles especially those of QTL with high *R*^2^ and relatively large contributions will be preferentially selected in breeding. In this study, *QTL.5.3* and *QTL.5.4* had the highest RC while *QTL.3.1* and *QTL.3.2* had the highest *R*^2^ values (Figure 3). These four QTL could be taken into consideration for future breeding. For those epistatic QTL, additive effects still contributed resistance to marsh spot disease. Therefore, in a breeding program, the epistatic effect of QTL markers should also be considered in the selection of molecular markers for optimal QTL combinations.

A total of 1273 SNP markers identified from the RIL population were used for QTL mapping in this study. The estimates of genomic heritability for marsh spot resistance indicate that more than half of the phenotypic variation can be explained using these SNPs, providing the potential to perform GS to improve marsh spot resistance in common bean breeding.

### 3.4. Candidate Gene Prediction

Although QTL mapping or GWAS have been widely used to identify QTL or QTNs associated with traits of importance and to predict potential candidate genes, their applications were limited. First of all, the prediction of candidate genes relies on whether the detected QTL/QTNs are not false positives. Then, the application of designating a potential candidate gene of a QTL depends on many other factors, such as the number of markers used, marker density on chromosomes, the recombination rate of genomic regions, and so on. To date, the most popular and simple approach for predicting candidate genes is to investigate the annotated genes in the vicinity of the QTL, such as a window of a specific physical distance flanking the QTL [51,52,53]. Although functional validation is the ultimate goal, candidate gene prediction of QTL/QTNs based on chromosomal location combined with *a priori* knowledge of gene functions has the potential to significantly narrow down the candidate gene list. This approach requires a list of annotated genes associated with the traits that have been validated to some extent in previous studies.

Marsh spot symptoms are likely caused by Mn deficiency due to the low availability of soil Mn, a limited capacity for Mn uptake and transport, and/or because of interference from other physiological pathways involving Mn, such as deoxidation [16,18,54,55,56,57]. Our previous study indirectly showed that Mn concentration in soil may be associated with the development of marsh spot in cranberry beans [19]. Here, candidate genes with the possible function of Mn transporter and deoxidation were annotated. Two Mn transporter protein-coding genes are co-located to *QTL 5.2*: one is the zinc transporter (ZIP) coding gene *Phvul.005G048900*, and the other one is the cation efflux (CAX) family protein-coding gene *Phvul.005G049300*. ZIP family members are involved in the transport of Mn in stellar root cells and present in the tonoplast, and took part in remobilizing Mn from the vacuoles to the cytoplasm [58]. The members of the CAX family are metal transporters and mainly control the influx of cations into the vacuole [59]. CAX-like transporters were found in other species, such as *LeCAX2* in tomato (*Solanum lycopersicum* L.), and *HvCAX2* in barley (*H. vulgare*). They are expressed ubiquitously in the roots, shoots, immature spikes and seeds [29].

Mn, naturally plentiful in most soils, should be adequately available to plants. However, deficiencies occur when in soils with high pH, high organic matter or during cold and wet conditions. Some of the identified candidate genes could play a role in Mn regulation in plants. Such as the ZIP gene mentioned above, the expression of the ZIP gene could allow plants to absorb more Mn from soil or remobilize more Mn to seeds during germination, thus, the development of marsh spot in seeds could be prevented or reduced.

## 4. Methods and Materials

### 4.1. Recombinant Inbred Lines (RILs)

An F_2:7_ RIL population of 138 individuals derived from a cross between the marsh spot susceptible cultivar “Messina” and the highly-resistant cultivar ‘Cran09’ was generated [19]. F_2_ plants were selfed and propagated by single seed descent to the F_7_ generation to ensure a high percentage (>98%) of homozygosity.

### 4.2. Phenotyping of Marsh Spot Resistance

From 2015 to 2019, the 138 RILs and their two parents were evaluated for marsh spot severity in sandy and heavy clay soils as previously described [19]. Briefly, the field trials were conducted in a partially balanced lattice design with three replications at the Morden Research and Development Centre, Morden, Manitoba, Canada (49°11′ N, 98°5′ W). Each line was planted in a 5 m-long row with 75 cm spacing between rows, and herbicides and fertilizers were applied to ensure optimal growth following standard commercial production guidelines. After harvest, ten seeds were randomly selected from each line and rated for marsh spot severity using a 0 to 5 scale, where 0 indicates no symptoms and 5 represents the most severe symptoms. The marsh spot resistance index (MSRI) was used to estimate the severity of the disease for each of the RILs:$$\mathrm{MSRI}=\frac{\sum_{i=0}^{n}\left(\mathrm{number}\;\mathrm{of}\;\mathrm{seeds}\;\mathrm{at}\;a\;\mathrm{rating}\;\mathrm{with}\;0-5\;\mathrm{scale}\;\times \;\mathrm{the}\;\mathrm{rating}\right)}{\mathrm{Total}\;\mathrm{number}\;\mathrm{of}\;\mathrm{seeds}}$$
where *n* is the total number of ratings and *i* = 0, 1, …, 5, respectively.

A total of 18 phenotypic datasets were collated: ten for each combination of the five years and two soil types, five for the means of each year over the soil types, two for the means of soil types over the five years and one for the means overall years and soil types. Statistical illustrations were drawn using the R package ‘ggplot2’ (https://cran.r-project.org/web/packages/ggplot2/index.html, (accessed on 1 May 2021)). The detailed ratings for marsh spot and the analyses of the phenotypic data were carried out as previously described [16,19].

### 4.3. Genotyping by Sequencing and SNP Identification

Seeds of the individual RILs along with the parental lines were grown in a growth chamber. At the 2-leaf stage, 75 mg of leaf tissue was sampled and flash-frozen in liquid nitrogen before being lyophilized in a FreeZone benchtop freeze-dryer (Labconco, Kansas City, MO, USA). Genomic DNA was extracted using the DNeasy 96-well kit (Qiagen, Germantown, MD, USA) and quantified with the Quant-iT™ PicoGreen™ dsDNA assay kit (ThermoFisher, Waltham, MA, USA) following the manufacturer’s instructions. The DNA samples were diluted to 20 ng/µL, and 10 µL of each sample was used for library construction.

The library preparation and sequencing service was provided by the Centre d’expertise et de services Génome Québec (Montréal, QC, Canada). The GBS library was constructed for each of the 138 RILs and ten libraries each for the parents, for a total of 158 libraries. Library construction was conducted at the Institute of Integrative Biology and System (IBIS, Université Laval, Québec, QC, Canada) using the MspI/PstI restriction enzyme combination as previously described [60]. The 158 indexed libraries were pooled and sequenced on 20 35 M-read NovaSeq 6000 lanes using the paired-end 150 bp (PE150) mode at the Centre d’expertise et de services Génome Québec.

As the cranberry common bean belongs to the Andean gene pool [61], the common bean reference genome v2.1 of Andean type landrace G19833 [31] was used as a reference for SNP discovery. The generated raw read data were filtered using the AGSNP pipeline [62] for standard quality and aligned to the reference genome using the Burrows-Wheeler Alignment tool (BWA V0.78-r455). Variant detection was performed using SAMTools V1.15.1 [63]. The entire procedure was implemented in the updated custom GBS analysis pipeline [64,65]. As a quality check, only SNPs that were polymorphic between parents and that weresegregated in the RIL population were selected. Then, SNPs with MAF > 0.01 and call rate > 20% were retained. Missing SNPs were then imputed using Beagle V5.1 [35] to produce the SNP dataset for linkage map construction. SNPs were assigned to LD blocks (D′ > 0.8) using the R package gpart V1.13.0 (http://bioconductor.org/packages/release/bioc/html/gpart.html, (accessed on 2 June 2021)) and one representative SNP was chosen to represent each block. Because the construction of a genetic linkage map and the detection of QTL may be influenced by segregation distortion, the Chi-square test in IciMapping V4.2 (https://isbreeding.caas.cn/rj/qtllcmapping/294445.htm, (accessed on 24 July 2019)) [66] was used to evaluate the significance of segregation ratios and SNPs that significantly deviate from the expected 1:1 ratio (*p* < 0.05) were excluded.

### 4.4. Genomic Heritability

Genomic heritability (*h*^2^), representing the proportion of additive genetic variance component of the total phenotypic variance, was estimated for all SNPs using the R package ‘sommer’ V4.1 (https://cran.r-project.org/web/packages/sommer/index.html, (accessed on 5 January 2021)) with the genomic best linear unbiased prediction (GBLUP) model.

### 4.5. Construction of Linkage Map

Construction of the linkage map was performed using QTL IciMapping V4.2 software [66]. The SNPs were divided into linkage groups based on their physical positions on chromosomes and subsequently ordered based on their recombinant frequencies. A maximum recombination frequency of 0.35 centimorgan (cm) was used from three criterion options. Genetic map distances were estimated using the Kosambi mapping function [67]. The linkage groups were assigned to their corresponding chromosomes based on the SNPs identified on the reference genome [31].

### 4.6. QTL Identification

IciMapping V4.0 [66] with the inclusive composite interval mapping (ICIM) and composite interval mapping (CIM) models was used for QTL mapping of the RIL population. In ICIM, forward and backward stepwise regressions were first computed, and Expectation–maximization (EM) iterations were then applied to consider all markers simultaneously. The additive (ICIM-ADD) and additive + epistatic (ICIM-EPI) models were used to detect QTL with additive and/or epistatic effects, respectively.

Genome-wide composite interval mapping (GCIM) [45] was also used to detect the large and small effects of QTL. The GCIM model includes two steps. The first involves the scanning of putative QTL across the genome using a single-locus random mixed linear model used in GWAS, and the second is the integration of the selected putative QTL into a multi-QTL mixed linear model. The QTL effects were calculated by the empirical Bayes method with the likelihood ratio test employed on true QTL detection [45]. GCIM was implemented using the R package QTL.gCIMapping.GUI V2.1.1 (https://cran.r-project.org/web/packages/QTL.gCIMapping.GUI/index.html, (accessed on 10 December 2021)).

Permutation tests of 1000 iterations [68] under the type I error *α* = 0.05 was performed to obtain the LOD scores to be used as thresholds of significance for QTL detection.

A haplotype block-based GWAS method, RTM-GWAS (restricted two-stage, multi-locus, multi-allele GWAS) V2020.0 [43], was also employed to detect QTL regions. RTM-GWAS first groups all SNP markers that shared strong linkage disequilibrium (LD) (D’ > 0.8) into LD blocks, and then uses those LD blocks for QTL detection. A significance level of 0.05 was used for the pre-selection of individual candidate markers under the single locus model, and an experiment-wide significance level of 0.05 was used for the stepwise regression to declare the significant QTL under the multi-locus model.

One way-ANOVA was performed for each QTL to further test the statistical significance of MSRI between QTL alleles in all 18 phenotypic datasets. QTL were considered as single fixed factors with two or more alleles. In addition, the *R*^2^ of each QTL was estimated as the proportion of phenotypic variation in the RIL population explained by alleles of the QTL. A higher *R*^2^ value indicates the QTL has a stronger effect on the marsh spot resistance. The average *R*^2^ of a QTL was calculated using *R*^2^ values that were statistically significant in all 18 datasets.

To calculate the relative contribution (RC) of each QTL to total phenotypic variation, the relative *R*^2^ value of each QTL was calculated based on a linear model containing all QTL using the R package ‘relaimpo’ V2.2 (https://cran.rstudio.com/web/packages/relaimpo/index.html, (accessed on 10 December 2021)). To evaluate the overall contribution of all detected QTL, the likelihood-ratio-based *R*^2^ between MSRI and all identified QTL in all 18 datasets were estimated using R package ‘MuMln’ V1.46.0 [32].

### 4.7. Favorable Alleles

To determine the number of favorable alleles of each RIL, the mean MSRI of individuals with the same alleles was calculated for each allele of the identified QTL. If a QTL spans more than two SNPs, the number of alleles present at a QTL may be greater than two. Theoretically, no recombination between SNPs within a QTL region is expected, and eventually, most QTL had only two alleles. The recombinant alleles had very low frequencies in the population. These recombinant alleles may be due to rare recombination events between SNPs or result from the errors of SNP imputation. Therefore, only two alleles with the highest frequencies were considered for each QTL. The allele with a high mean MSRI value was assigned a favorable allele, whereas another allele with a low mean MSRI value was assigned an unfavorable allele. For each of the RILs, the total number of favorable alleles for all QTL were counted.

### 4.8. Candidate Gene Prediction

To predict the candidate genes associated with marsh spot disease resistance, a list of gene families related to Mn transport, Mn efficiency or Mn content in plants was compiled (Appendix A) [16] based on the common bean reference genome sequence [31]. The protein sequences of Mn-deficiency-related gene families from common bean, *Arabidopsis*, rice (*Oryza sativa*) and barley (*Hordeum vulgare* L.) (https://www.ncbi.nlm.nih.gov/guide/proteins/, (accessed on 1 May 2021)) were extracted and BLAST (basic alignment search tool) [69] searches were performed against the common bean reference genome sequence to locate the bean orthologous sequences. A total of 154 candidate genes were eventually identified, belonging to ten gene families. A genome-wide scan was performed to identify the ones located within 100 Kb of QTL to constitute the list of candidate genes co-located with the identified QTL.

## 5. Conclusions

QTL mapping in this study determined that marsh spot resistance in cranberry common bean is a highly heritable trait that is genetically controlled by multiple genes. Although four gene loci with additive and epistatic effects have been detected through joint segregation analysis at a phenotype level as reported previously [19], sixteen additive and three pairs of epistatic QTL were further identified via QTL mapping at the genomic level in this study. These QTL explained up to 81% of phenotypic variation. Despite epistatic effects between some QTL, there existed a significant correlation between the number of favorable alleles of the additive QTLs and marsh spot resistance (*R*^2^ = 72%), which confirmed that the favorable alleles of these QTL are additive and can be pyramided in future common bean cultivars by MAS. These QTL will facilitate the development of molecular markers for resistance breeding. In addition, 13 candidate genes related to Mn deficiency or Mn content in plants were shown to be co-located within six QTL regions. Those genes will be further validated in future functional genomic studies to determine their potential to improve marsh spot resistance in germplasm or new cultivars by adopting modern genetic improvement techniques such as genome or gene editing.

## Figures and Tables

**Figure 1 ijms-23-07639-f001:**
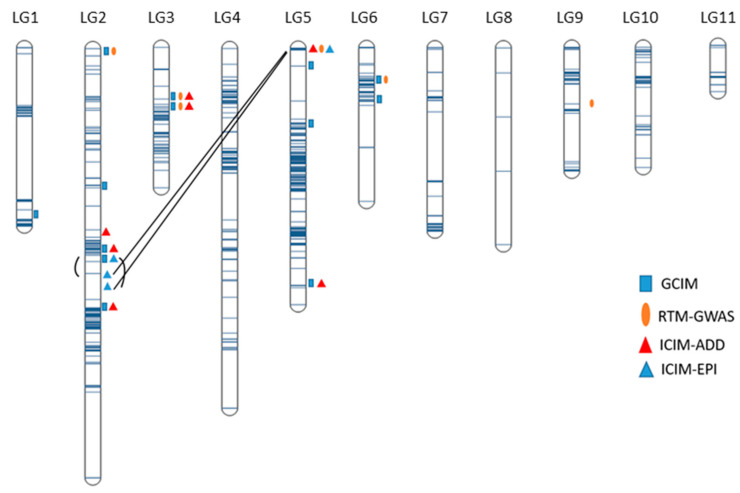
Distribution of 1273 single nucleotide polymorphisms (SNPs) in the genetic map and quantitative trait loci (QTL) associated with marsh spot resistance index (MSRI) identified using four statistical models represented in the linkage map. Two QTL with significant epistasis effects are linked by a line or a curve.

**Figure 2 ijms-23-07639-f002:**
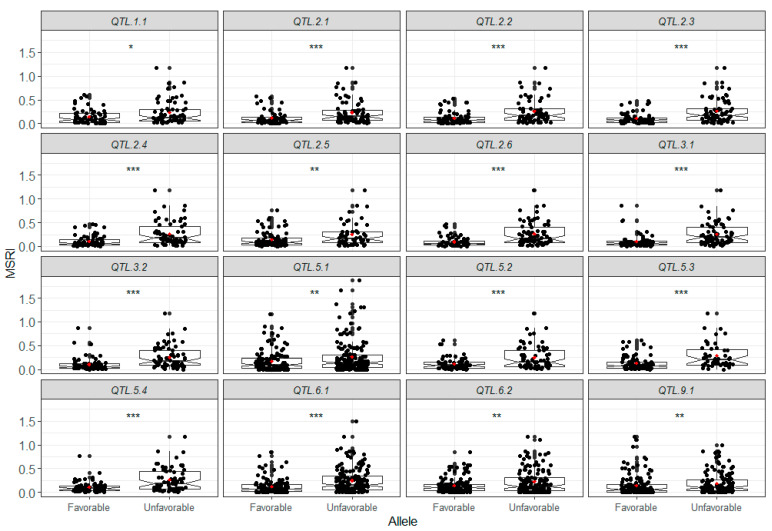
Box plots of 16 quantitative trait loci (QTL) associated with marsh spot resistance index (MSRI). *: *p* < 0.05; **: *p* < 0.01; ***: *p* < 0.0001. In each box plot, the black dots represent data points and the red dot represents the mean of the data points.

**Figure 3 ijms-23-07639-f003:**
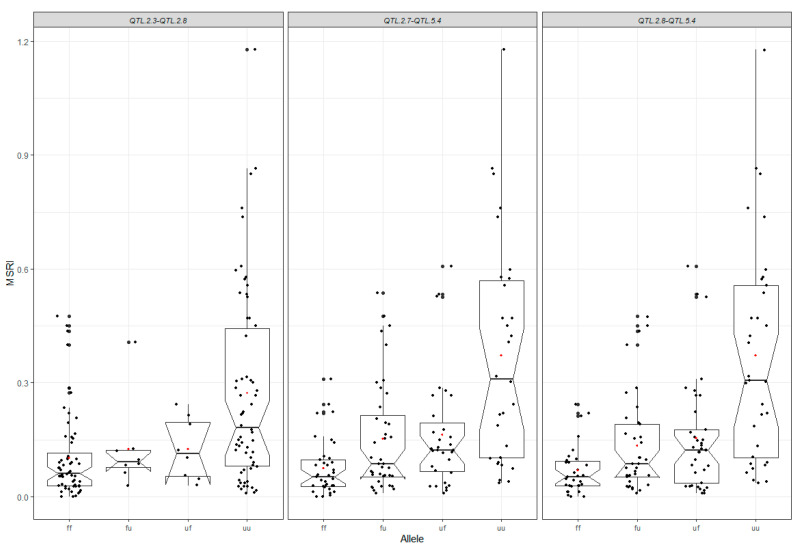
Box plots of marsh spot resistance index (MSRI) in terms of the combinations of favorable alleles of three quantitative trait locus (QTL) pairs identified using the ICIM-EPI model. In each box plot, the black dots represent data points and the red dot represents the mean of the data points. ff: both QTL 1 and QTL 2 were favorable alleles. uf: QTL 1 was an unfavorable allele while QTL 2 was a favorable allele; fu: QTL 1 was a favorable allele but QTL 2 was an unfavorable allele; uu: both QTL 1 and QTL 2 were unfavorable alleles.

**Figure 4 ijms-23-07639-f004:**
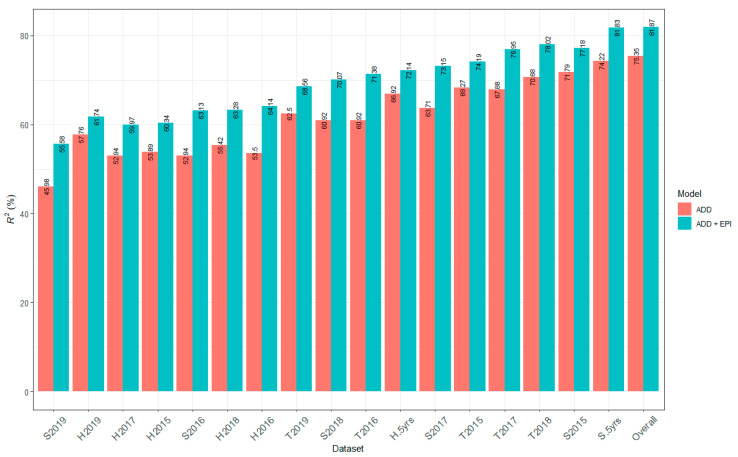
Bar chart of *R*^2^ of 16 additive (ADD) and all 18 additive and epistatic (ADD+EPI) quantitative trait loci (QTL) to the 18 phenotypic datasets of marsh spot resistance index (MSRI). The likelihood-ratio-based *R*^2^ value for each phenotypic data set was calculated as the coefficient of determination in the multiple regression of 16 or 18 QTL on the phenotypic data set using R package MuMln V1.46.0 [32].

**Figure 5 ijms-23-07639-f005:**
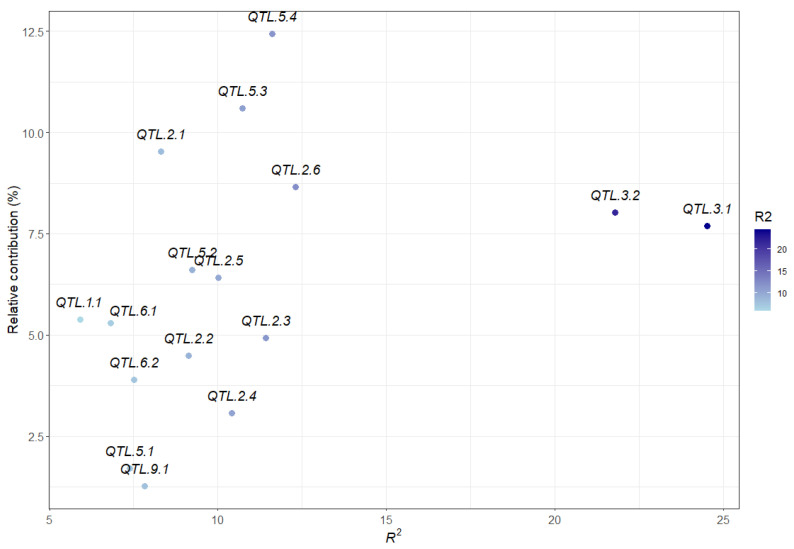
Relationship between the relative contributions (%) and mean *R*^2^ of 16 quantitative trait loci (QTL).

**Figure 6 ijms-23-07639-f006:**
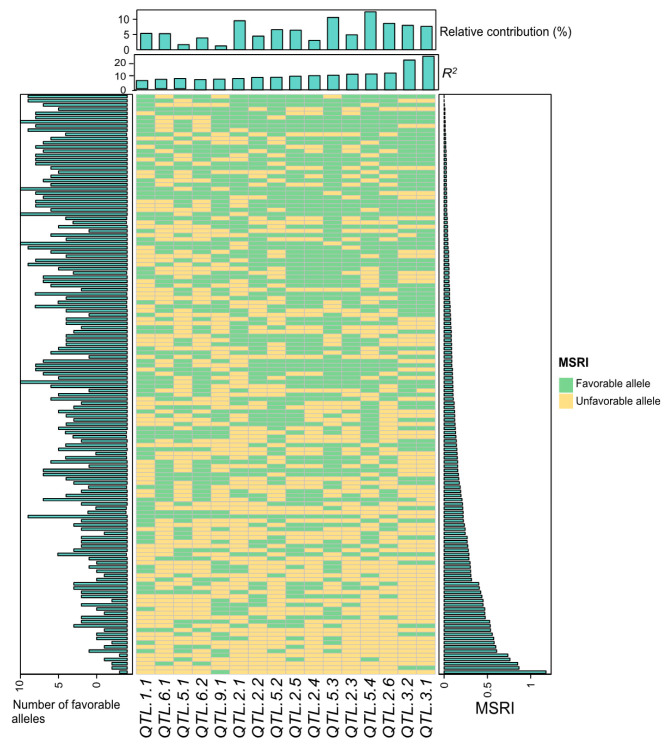
Favorable alleles of 16 quantitative trait loci (QTL) for marsh spot resistance index (MSRI) in the 138 recombinant inbred lines (RILs). The mean dataset over five years and two soil types was used. *R*^2^ refers to phenotypic variation (%) in MSRI explained by a QTL.

**Figure 7 ijms-23-07639-f007:**
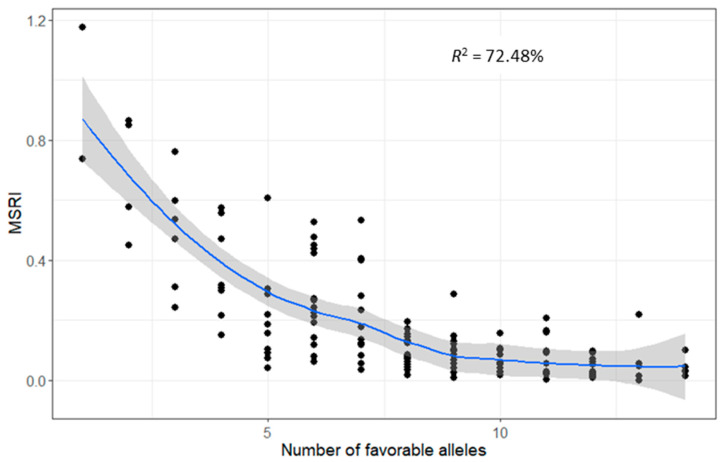
Relationship between marsh spot resistance index (MSRI) and the number of favorable alleles in the 138 recombinant inbred lines (RILs). The confidence intervals were drawn with a confidence of 0.99. The data of the overall means over years and soil types were used for calculation.

**Figure 8 ijms-23-07639-f008:**
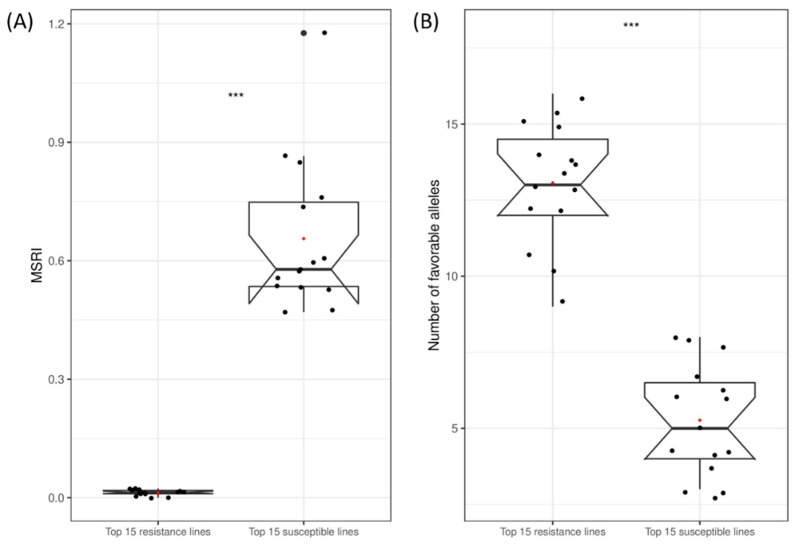
Boxplots of marsh spot resistance index (MSRI) (**A**) and the number of favorable alleles (**B**) for the 15 most resistant lines and the 15 most susceptible lines. ***: *p* < 0.0001. In each box plot, the black dots represent data points and the red dot represents the mean of the data points.

**Table 1 ijms-23-07639-t001:** Number of markers on the linkage groups (chromosomes).

Linage Group (Chromosome)	No. of Markers	Total Distance (cm)	No. of Recombination Intervals	$\bar{x}\pm s\;(\mathrm{cm})$
1	101	131	27	4.86 ± 13.40
2	263	305	96	3.18 ± 7.17
3	115	105	41	2.56 ± 3.75
4	160	257	69	3.73 ± 7.10
5	360	186	76	2.44 ± 4.39
6	71	114	28	4.08 ± 9.60
7	72	135	21	6.41 ± 11.52
8	9	144	4	36.11 ± 14.80
9	59	93	26	3.59 ± 7.73
10	44	91	26	3.50 ± 5.40
11	19	38	9	4.20 ± 6.32
Total	1273	1599	434	3.78 ± 8.13

$\bar{x}\pm s$: mean genetic distance between markers ± standard deviation (cm).

**Table 2 ijms-23-07639-t002:** Genomic heritability (*h*^2^ ± *s*) of the marsh spot resistance index (MSRI) of 138 recombinant inbred lines (RILs) from the Cran09/Messina population.

Phenotypic Dataset	Genomic Heritability (*h*^2^ ± *s*) (%)	Phenotypic Dataset	Genomic Heritability (*h*^2^ ± *s*) (%)
H2015	24.22 ± 0.09	S2019	12.07 ± 0.07
H2016	18.43 ± 0.08	T2015	32.48 ± 0.10
H2017	32.96 ± 0.10	T2016	28.02 ± 0.01
H2018	33.13 ± 0.10	T2017	45.53 ± 0.11
H2019	32.82 ± 0.10	T2018	45.48 ± 0.10
S2015	27.03 ± 0.10	T2019	41.12 ± 0.11
S2016	24.76 ± 0.10	H-5 yrs	46.46 ± 0.11
S2017	30.37 ± 0.10	S-5 yrs	47.14 ± 0.11
S2018	16.67 ± 0.08	Overall	55.91 ± 0.10

H: heavy clay soil; S: sandy soil; T: means of years over two soil types; H-5 yrs: means of heavy clay soil over five years; S-5 yrs: means of sandy soil over five years; Overall: means over five years and two soil types.

**Table 3 ijms-23-07639-t003:** Sixteen quantitative trait loci (QTL) detected from the recombinant inbred line (RIL) population of 138 individuals.

QTL	Flanking Markers and Position	LG Pos (cm)	Additive Effect	No. of Datasets ^(a)^	Significant Datasets ^(b)^	Average *R*^2^ (%) ^(c)^	Model
*QTL.1.1*	Chr1_48339634–Chr1_50146614	119.26–126.33	0.10	1	3	5.92	GCIM
*QTL.2.1*	Chr2_872663–Chr2_1135128	0.06–5.22	0.05–0.11	6	11	8.30	GCIM, RTM-GWAS
*QTL.2.2*	Chr2_32113326	97.46	0.07	1	17	9.12	GCIM
*QTL.2.3*	Chr2_34070996–Chr2_35065692	147.11–151.45	0.15	1	16	11.43	GCIM
*QTL.2.4*	Chr2_35130486–Chr2_35289581	143.04–142.69	0.07–0.10	5	18	10.42	GCIM, ICIM-ADD
*QTL.2.5*	Chr2_35344261–Chr2_36750706	128.94–134.16	0.14	1	17	10.02	ICIM-ADD
*QTL.2.6*	Chr2_37937595–Chr2_38452857	187.14–188.25	0.05–0.51	5	18	12.30	ICIM-ADD, GCIM
*QTL.3.1*	Chr3_11944447–Chr3_19043093	47.42–51.49	−0.08	5	17	24.52	GCIM, RTM-GWAS, ICIM-ADD
*QTL.3.2*	Chr3_19701297–Chr3_30221015	48.9–50.4	−0.57–0.17	13	17	21.78	ICIM-ADD, GCIM, RTM-GWAS
*QTL.5.1*	Chr5_11498360–Chr5_19238819	56.2–57.31	−0.46	1	4	7.39	GCIM
*QTL.5.2*	Chr5_1647320–Chr5_31681432	12.94–38.3	0.06–0.10	6	18	9.23	GCIM
*QTL.5.3*	Chr5_38536162–Chr5_38536272	171.77–171.75	0.06–0.13	10	17	10.72	GCIM, ICIM-ADD
*QTL.5.4*	Chr5_623370–Chr5_673021	0–1.13	−0.10–−0.07	12	17	11.61	ICIM-ADD, RTM-GWAS
*QTL.6.1*	Chr6_1374720–Chr6_1504675	23.18–23.92	0.04–0.08	4	13	6.81	GCIM, RTM-GWAS
*QTL.6.2*	Chr6_13598278–Chr6_14124318	39.58–38.48	0.05–0.07	2	5	7.52	GCIM
*QTL.9.1*	Chr9_17827630–Chr9_20865151	42.84–46.75	0.20	2	3	7.82	GCIM, RTM-GWAS

^(a)^ Number of datasets where QTL detected from; ^(b)^ number of datasets which QTL significantly correlated with; ^(c)^ the mean of *R*^2^ of QTL in the datasets that showed significant correlation with QTL; Chr: chromosome; LG pos: position on linkage group.

**Table 4 ijms-23-07639-t004:** Quantitative trait loci (QTL) detected using the additive-epistatic model ICIM-EPI.

	QTL 1				QTL 2			No. Datasets with QTL ^(a)^	No. Datasets with Significant Effect ^(b)^	Average of *R*^2^ (%) ^(c)^	Additive Effect of QTL 1	Additive Effect of QTL 2	Epistatic Effect
QTL 1	Left Marker	Right Marker	LG Pos (cm)	QTL 2	Left Marker	Right Marker	LG Pos (cm)						
*QTL.5.4*	Chr5_673021	Chr5_1647342	1.13–12.92	*QTL.2.7*	Chr2_36996368	Chr2_37937763	160.01–178.45	1	18	30.64	−0.05	0.04	−0.11
*QTL.5.4*	Chr5_673021	Chr5_1647342	1.13–12.92	*QTL.2.8*	Chr2_37937763	Chr2_37531627	178.45–184.57	2	18	30.69	−0.05	0.04	−0.11
*QTL.2.3*	Chr2_34070996	Chr2_35065692	147.11–151.45	*QTL.2.8*	Chr2_37937763	Chr2_37531627	178.45–184.57	1	9	16.31	−0.06	0.04	−0.10

^(a)^ Number of datasets where QTL detected from; ^(b)^ number of datasets which QTL significant correlated with; ^(c)^ the mean of average *R*^2^ of two QTL. Chr: chromosome; LG pos: position on linkage group.

**Table 5 ijms-23-07639-t005:** Candidate genes co-located in the regions of the quantitative trait loci (QTL).

QTL	Gene	Chr	Gene Coordinates ^(a)^	Annotation
*QTL.1.1*	*Phvul.001G250300*	1	50100939–50102262	Heavy metal transport/detoxification superfamily protein
	*Phvul.001G247400*	1	49894723–49895275	Heavy metal transport/detoxification superfamily protein
*QTL.2.3*	*Phvul.002G184200*	2	34468080–34481795	ZIP metal ion transporter family
*QTL.3.1*	*Phvul.003G086300*	3	16877488–16879137	Heavy metal transport/detoxification superfamily protein
*QTL.3.2*	*Phvul.003G092500*	3	21113140–21115717	Heavy metal transport/detoxification superfamily protein
	*Phvul.003G104900*	3	23350962–23352673	Heavy metal transport/detoxification superfamily protein
	*Phvul.003G099700*	3	25708027–25708491	Heavy metal transport/detoxification superfamily protein
	*Phvul.003G108900*	3	27536901–27538128	Heavy metal transport/detoxification superfamily protein
*QTL.5.2*	*Phvul.005G095400*	5	29797811–29799135	Heavy metal transport/detoxification superfamily protein
	*Phvul.005G049300*	5	5742223–5745905	Cation efflux family protein
	*Phvul.005G048900*	5	5682746–5684903	Zinc transporter 1 precursor
*QTL.9.1*	*Phvul.009G137100*	9	20653368–20662253	Manganese tracking factor for mitochondrial SOD2
	*Phvul.009G127900*	9	19429094–19433403	NRAMP metal ion transporter 6

^(a)^ All the candidate genes are located within the QTL regions. Chr: chromosome.

## Data Availability

The raw sequencing data are deposited in the NCBI database under the BioProject accession number PRJNA827136 with SRA accession numbers SRR18791987-SRR18792037. The SNP data (vcf file) is deposited in Zenodo at https://zenodo.org/record/6448295#.Yl63bllE0-J, (accessed on 11 April 2022).

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
