# Peer review of "Quantitative Trait Locus Mapping of Marsh Spot Disease Resistance in Cranberry Common Bean (Phaseolus vulgaris L.)"

_ijms, 2022, doi:10.3390/ijms23147639_

Round 1

Reviewer 1 Report

The authors identified QTL associated with resistance to Marsh spot in common bean. Comments follow below.

1. If Mn deficiency is associated with the symptom, levels of Mn in each environment should affect the results. Please provide any data that can address amount of Mn in soils.

2. One major flaw is genetic map, which is very significant in genetic mapping. The length of genetic map was too lengthy compared to other common bean genetic maps in previous studies. Looking at the supplementary file, it is strongly suspected that the genetic map might be constructed incorrectly. Chromosome 1 is about 664 cM in the present study, which is not a reasonable number. I suggest that some markers should be cleaned up and removed if their genetic position is unreasonable or untrustworthy. The authors should double-check correlation between physical and genetic positions, and this should be provided as a supplementary data to demonstrate the relevance of genetic map.

3. Some QTL, such as QTL3.3 and QTL5.1, are not significant in Figure2. Why were they included in the figure?

4. After refining the genetic map and reconducting QTL analysis, the list of significant QTL may be changed.  

5. For better interpretation of epistatic interaction (Figure4), low significant QTL or low R2 QTL should be ignored.

6. Please double check the supplementary files. There are mismatches found in the titles and file numbers.

Reviewer 2 Report

The paper titled “Quantitative Trait Locus Mapping of Marsh spot Disease Resistance in Cranberry Common Bean (Phaseolus vulgaris L.)” presented a well-elaborated research. The authors performed a genetic mapping for the marsh spot resistance in Cranberry common bean by using138 recombinant inbred lines. Of which, 21 quantitative trait loci (QTL) associated with marsh spot resistance were identified, and 17 Mn-deficiency related candidate genes co-located within six QTL. Generally, I believe that the Author(s) provided solid theoretical foundations for the analysis using appropriate approaches. But there are still some errors need to be revised in the article, and my main concerns are listed as following:

Point 1: Is there conclusive evidence that the marsh spots are caused by manganese (Mn) deficiency and not other elements during seed development? If it is, the authors should provide more reliable references or robust experiments to confirm it.

Point 2: the zinc transporter (ZIP) coding gene Phvul.005G048900 and the cation efflux (CAX) family protein-coding gene Phvul.005G049300 were considered two important Mn transporter genes. The author should provide accurate reference information in introduction.

Point 3: What's the difference between Dry common bean and Cranberry Common Bean?

Point 4: The “195SNPs” should change to “195 SNPs” in line 97. And mistakes same as this should be changed in line 98,105,130 and so on.

Point 5: All of the tables in your paper should be changed to the standard three-line table.

Point 6: The quotation of database should be traced back to articles in which uploaded it. Please check on your reference format carefully and modify it.

Point 7: The “thus” is recommended to change to “eventually” in line 552.

Round 2

Reviewer 1 Report

The manuscript was improved and the authors appropriately responded to the previous comments.